# Effect of Summer Sunshine Exposure on Vitamin D Status in Young and Middle Age Poles: Is 30 ng/mL Vitamin D Cut-Off Really Suitable for the Polish Population?

**DOI:** 10.3390/ijerph18158116

**Published:** 2021-07-31

**Authors:** Magdalena Basińska-Lewandowska, Andrzej Lewiński, Wojciech Horzelski, Elżbieta Skowrońska-Jóźwiak

**Affiliations:** 1“Your Family Doctor”, General Practice Surgery, 93-324 Lodz, Poland; mblbas@poczta.onet.pl; 2Department of Endocrinology and Metabolic Diseases, Polish Mother’s Memorial Hospital—Research Institute, 93-338 Lodz, Poland; andrzej.lewinski@umed.lodz.pl; 3Department of Endocrinology and Metabolic Diseases, Medical University of Lodz, 90-419 Lodz, Poland; 4Department of Mathematics and Computer Science, University of Lodz, 90-238 Lodz, Poland; wojciech.horzelskil@wmii.uni.lodz.pl

**Keywords:** vitamin D deficiency, vitamin D insufficiency, optimal cut-off, prevalence, healthy population

## Abstract

Background: There is no consensus regarding vitamin sufficiency status with either 20 ng/mL or 30 ng/mL sufficiency cut-off. We assessed the effects of summer sunshine exposure on vitamin D status. Participants: We measured vitamin D concentrations, PTH, creatinine, and total calcium in 132 healthy subjects, age 29.36 ± 13.57 years, in spring and autumn. Results: There was an overall increase in vitamin D concentrations from spring to autumn from 18.1 ± 7.39 ng/mL to 24.58 ± 7.72 ng/mL, (*p* < 0.001), accompanied by a decrease in PTH from 44.4 ± 17.76 pg/mL to 36.6 ± 14.84 pg/mL, (*p* < 0.001). In spring, only 5.3% of individuals were vitamin D sufficient for a 30 ng/mL cut-off, increasing to 23.2% in autumn (*p* < 0.001). In contrast, when a 20 ng/mL cut-off was employed, vitamin D sufficiency was found in 34.1% in spring and 66.4% individuals in autumn, respectively, (*p* < 0.001). In multiple regression analysis, holiday leave was the only significant determinant of vitamin D increase (*p* < 0.001). Conclusions: Holiday leave is the strongest determinant of an increase in vitamin D. The majority of healthy individuals fail to reach a 30 ng/mL vitamin D cut-off after summer sunshine exposure. This raises the question, whether such a cut-off is indeed suitable for the Polish population.

## 1. Introduction

Though vitamin D deficiency and/or hypovitaminosis appears to be highly prevalent among various populations [1,2,3,4], there is no universal consensus pertaining to vitamin D reference ranges, and this is reflected in conflicting opinions between the position of the Endocrine Society [5] and Institute of Medicine (IOM) [6,7]. In particular, the Endocrine Society defines vitamin D deficiency as concentrations below 20 ng/mL (50 nmol/L) and vitamin D insufficiency for concentrations between 21–29 ng/mL (52.5–72.5 nmol/L). Still, IOM strongly argues that vitamin D sufficiency applies to concentrations above 20 ng/mL (50 nmol/L), with vitamin D deficiency being only present for concentrations below 12 ng/mL (30 nmol/L) [8]. 

Furthermore, there are significant differences in vitamin D concentrations depending on the season of the year (spring vs. autumn) [9,10]. Regardless of the accepted reference range, it appears that vitamin D deficiency/insufficiency appears to be most prevalent in younger subjects as older individuals are more likely to take vitamin D supplements for osteoporosis prevention, particularly in the case of females [11]. For instance, in the Polish population, the lowest vitamin D concentrations were noted in the 10–15-year-old group, while the highest was noted in those aged 70–80. Similar data were observed from Canada [12], where the lowest vitamin D concentrations were observed in a group of 20–39 years old. 

On the strength of that, we have endeavoured to assess vitamin D status in young and middle-aged healthy individuals in early spring and after the summer period to ascertain whether summer-related sunshine exposure, without any extra supplementation, would be enough to ensure a vitamin D sufficiency status depending on applied deficiency cut-off (i.e., 20 ng/mL vs. 30 ng/mL). 

## 2. Materials and Methods

### 2.1. Study Procedure and Participants

The study included 132 subjects (56 males, 76 females) age 29.36 ± 13.57 (mean ± SD), range 6–50 years, BMI 25.32 ± 6.33 kg/m^2^. There were no sex-related differences in BMI (25.06 ± 6.12 kg/m^2^ versus 25.53 ± 6.52 kg/m^2^) or age (26.72 ± 13.36 years versus 30.41 ± 14.58 years, for males and females, respectively). We have decided to select this particular age range (i.e., 6–50 years), as these individuals rarely take any vitamin supplements outside the pregnancy period. At the same time, our study was designed to assess the results of physiological sunshine exposure and not the effects of supplementation. On the other hand, women above 50 often take vitamin D supplements for osteoporosis prevention, while pre-school children also often take vitamin D supplements. After informed consent, concentrations of 25-OH-vitamin D, parathormone (PTH), and total calcium were assessed in late winter/early spring (March–April) and late summer/early autumn (September–October). Repeated measurements were obtained in 125 subjects. All subjects had normal kidney function, had no history of liver disease, were not vegetarians, and they were not taking any vitamin D supplements. Pregnant or breastfeeding women were not included in the study. All subjects were also free of any major disabilities that could impede their stay outdoors. Recommended daily vitamin D intake is at least 10 µg/day [13]. Unfortunately, we could not get valid data on average daily vitamin D ingestion for the Polish population in this age group. However, data from the UK (age range 12–60 years) suggest that average daily intake was lower with a median as low as 4 µg/day [14]. After informed consent, all investigated individuals were recruited from a single general practice in the city of Lodz (Poland).

25-hydroxy-vitamin D was measured by means of Elecsys Vitamin D Total II assay, using Cobas 801 analyser with intra-assay variation 1.1–3.1% and inter-assay variation 2.2–4.3%, while 1–84 amino acid PTH was measured by electrochemiluminescence “ECLIA” method – Elecsys system, Cobas 801 analyser with intra-assay variation 1.7–2.6%, inter-assay variation 2.5–3.1%. The reference range for PTH, according to this assay, is 15–65 pg/mL.

Vitamin D status was defined as follows:

Definition one: Vitamin D sufficiency > 20 ng/mL (50 nmol/L), insufficiency 12–20 ng/mL (30–50 nmol/L), vitamin D deficiency < 12 ng/mL (30 nmol/L) [6,7,8,15].

Definition two: Vitamin D sufficiency > 30 ng/mL (75 nmol/L), insufficiency 20–30 ng/mL (50–75 nmol/L), vitamin D deficiency < 20 ng/mL (50 nmol/L) [5].

Vitamin D concentrations below 10 ng/mL (25 nmol/L) were classified as “severe vitamin D deficiency” and are considered to be associated with the risk of osteomalacia [5,15].

### 2.2. Statistical Analysis

The MedCalc 19.0.7 package (MedCalc Software, Ostend, Belgium) was used for statistical analysis. Shapiro–Wilk and D’Agostino–Pearson tests were used to test the normality of distributions. The Student t-test and Mann–Whitney methods were used to compare parameters (after applying the Fisher-Snedecor test). Variance analysis (ANOVA) was used to compare more parameters. Changes in the distribution of analysed parameters were performed using the chi^2^ test. *p* < 0.05 ratio was taken as statistically significant. 

## 3. Results

There was an increase in vitamin D concentrations from spring to autumn from 18.1 ± 7.37 ng/mL (median 17.0 ng/mL) to 24.58 ± 7.72, ng/mL (median 23.1 ng/mL), *p* < 0.0001, Figure 1, and a decline in PTH concentrations from 44.40 ± 17.76 pg/mL (median 41.65 pg/mL) to 36.63 ± 14.84 pg/mL (median 33.50 pg/mL), *p* < 0.0001, Figure 2, without significant change in Calcium concentrations 9.53 ± 0.41 mg/dL vs 9.49 ± 0.76 mg/dL, *p* = 0.65. 

Vitamin D concentrations were higher in spring in women (19.38 ± 7.60 ng/mL (median 19.0 ng/mL) vs. 16.59 ± 6.82 ng/mL (median 15.1 ng/mL), *p* = 0.02), but these differences ceased to be significant in autumn, i.e., 25.37 ± 8.10 ng/mL (median 24.15 ng/mL) vs. 23.81 ± 7.11 ng/mL (median 23.11 ng/mL), *p* = 0.24. The summer-related increase in vitamin D concentrations was significant for both sexes (*p* < 0.001). There were no significant differences in PTH and calcium concentrations in men vs. women, while summer-related decrease in PTH concentrations was significant for both sexes (45.46 ± 17.59 pg/mL vs. 37.21 ± 15.07 pg/mL and 43.20 ± 18.16 pg/mL vs. 36.01 ± 13.88 pg/mL, *p* = 0.001, and *p* = 0.011, for women and men, respectively. 

Prevalence of vitamin D sufficiency, insufficiency, and frank deficiency depending on the accepted cut-offs is presented in Table 1. 

Season-related differences in the prevalence of vitamin D sufficiency, insufficiency, deficiency, and severe deficiency (<10 ng/mL) were highly significant (chi^2^ test, *p* < 0.001).

Correlation analysis between the analysed parameters is presented in Table 2.

Notably, the correlation between total calcium concentrations and PTH was significant only in men, while the correlation between PTH and vitamin D failed to reach statistical significance. 

There was a weak (r = −0.2) but significant correlation between summer-related vitamin D increment and BMI (*p* = 0.028). When the analysed population was divided into quartiles, the summer-related increments in vitamin D were not related to age (*p* = 0.23). However, they were still related to BMI with the highest increment in the lowest BMI quartile (13–21 kg/m^2^) and the lowest in the 3rd BMI quartile (24.6–28.0 kg/m^2^), *p* = 0.027. The highest vitamin D increments were observed in subjects who had at least two weeks of holiday leave, where we defined holiday leave as a period of at least two weeks spent outside their permanent place of residence, *p* = 0.005 (Figure 3).

The above-mentioned holiday-related increases in vitamin D concentrations were significant for both sexes (*p* = 0.0086 and *p* = 0.0089, for women, and men, respectively), with no significant difference in the degree of vitamin D increments between men and women (*p* = 0.13). It should be mentioned that given the concurrent COVID-19 pandemic, in a great majority of cases, holidays were spent in Poland, with only one person traveling to Croatia and one to Norway. When two week holiday period was put into a multivariate analysis, this turned out to be the only factor associated with increasing vitamin D concentrations (Table 3), while both age and BMI were no longer statistically significant. 

## 4. Discussion

Our study demonstrates that despite an expected increase in vitamin D concentrations and a fall in PTH after a summer period, most healthy individuals fail to reach a 30 ng/mL sufficiency cut-off. Though it could be argued that such a situation might stem predominantly from Poland’s location at a relatively high latitude, this appears not to be the case.

For instance, a study on a large number of individuals (*n* = 8183) in Crete [16], i.e., in a country with abundant sunshine and a population prone to outdoor activities, revealed a mean vitamin concentration of 19.48 ± 9.51 ng/mL in males and 18.01 ± 9.01 ng/mL in females, with peak mean vitamin D concentrations (August-September) around 28 ng/mL for men and 25 ng/mL, for women, for subjects up to the age of 50. Furthermore, a decrease in PTH concentrations was no longer significant for vitamin D concentrations above 20 ng/mL. Interestingly in the above study, in contrast to our data, vitamin D concentrations were higher in men, while in our study, women had higher vitamin D levels in spring. Still, this sex-related difference ceased to be significant in autumn. 

Suppose a majority of the population (and in our case, this applies to the cohort of individuals without any disabling diseases) fails to reach a pre-defined cut-off. In that case, this raises a question about the validity of this cut-off, i.e., in other words, whether such a cut-off is indeed suitable for this population. 

The typical procedure of determining the reference range relies on testing healthy volunteers and expressing obtained results either as mean ± 2SD or interquartile ranges. In the case of vitamin D, such a procedure may not be entirely appropriate due to the phenomenon of worldwide insufficiency. This is one of the reasons why Endocrine Society guidelines defined sufficiency of vitamin D as a level higher than 30 ng/mL based nadir of serum PTH concentrations in the setting of optimal intestinal calcium absorption. Yet, several data support the notion that vitamin D [i.e., 25(OH)D] concentrations above 20 ng/mL should be regarded as vitamin D sufficiency status. There are conflicting data regarding the relationship between the PTH plateau and 25(OH)D levels. For instance, Chapuy et al.[17], Thomas et al.[18] and Holick et al.[5] suggest that the PTH plateau occurs at vitamin concentrations between 30–40 ng/mL. However, Malabanan et al. [19] demonstrated that after administering 50 000 IU of vitamin D once weekly for 8 weeks the expected decrease in PTH was present only in subjects with a baseline vitamin D level of less than 20 ng/mL. Furthermore, intestinal calcium absorption seems to be related more to 1.25(OH)_2_vitamin D concentrations, rather to 25(OH) vitamin D [20], and no increase in intestinal calcium absorption can be observed after vitamin D supplementation when the baseline vitamin D level exceeds 15 ng/mL. In that context, it is worthwhile to note that increasing serum 25(OH)D levels from 22 to 40 ng/mL by daily administration of 2000 IU of vitamin D_3_ also did not increase serum 1.25(OH)2D concentrations [21]. Indeed, vitamin D and calcium supplementation did not increase serum 1.25(OH)_2_D levels, when 25(OH) vitamin D concentration was above 15 ng/mL [22]. Moreover, Sai et al. [23] suggest that vitamin D insufficiency should be defined as serum 25(OH)D less than 20 ng/mL as it relates to the bone. Though these data pertain to older subjects, it is also worthwhile to mention that a randomised trial of vitamin D supplementation (400, 800 or 1600 IU daily) failed to demonstrate differences in increments of bone mineral density [24]. Initial vitamin concentrations in this study were around 16 ng/mL in all subgroups, while daily supplementation with 400 IU of vitamin D was enough to obtain a mean vitamin concentration of 22.36 ng/mL vs 31.6 ng/mL a daily dose of 1600 IU. There are also doubts regarding the methodology of studies, where a 30 ng/mL vitamin D cut-off was proposed. In this respect, Priemel et al. [25] performed bone histomorphometry in 675 German individuals who died suddenly. The authors concluded that serum vitamin D levels above 30 ng/mL avoid osteoid excess and therefore defined this as the lowest threshold for bone health. Yet, this study was criticised for the authors overestimated the prevalence of osteomalacia by choosing an upper reference limit for osteoid volume at 2%, whereas 5% is usually quoted. Using this threshold (i.e., 5%), no osteoid excess was observed if serum 25(OH)D was above 20 ng/mL [20].

Despite these data, a 30 ng/mL vitamin sufficiency cut-off is universally quoted as desired reference range on laboratory results of vitamin D measurements, which clearly gives an impression to the majority of healthy individuals that they are vitamin “deficient” or at least “insufficient.” Furthermore, in 2014 vitamin D levels were the fifth most common laboratory test ordered for USA Medicare patients, with a total cost of $323 million [26]. Meanwhile, currently used methods to estimate bioavailable vitamin D do not consider the estimates of vitamin D binding proteins. What is more important, the threshold of 30 ng/mL encourages prescribing high-dose vitamin D preparations, which may be dangerous since several recent studies show a greater risk of falls [27] or respiratory infections after treatment with high-dose vitamin D [28,29,30]. Though issues of differences in 25(OH)D concentrations due to the use of different assays have been raised [31], this does not apply to our study due to the cross-over nature of the study design and the use of the same assay both in spring and in autumn. 

In our population, we failed to notice the significant relationship between BMI and vitamin D concentrations in women (r = 0.0279), with a significant negative correlation observed only in men (r = −0.339). A negative correlation between vitamin D concentrations and BMI was noted in women with polycystic ovary syndrome (PCOS) [32], and vitamin D deficiency seems to be more prevalent among obese individuals [33]. Our study has not performed a formal assessment of PCOS status, with relatively few women showing extreme obesity (mean BMI 25.53 kg/m^2^) with a 90th percentile BMI of 34.85 kg/m^2^. Notably, we observed a rather weak correlation between vitamin D and PTH concentrations (r = −0.208), similar to El Sabeh et al.[34], where coefficients of correlation between PTH and total, bioavailable and free vitamin D ranged from −0.22 to −0.25. It appeared, however, that neither age nor BMI, but at least a two week of holiday leave was the only significant variable that predicted a significant increase in vitamin concentrations after a summer period. Despite the ongoing COVID-19 pandemic, this phenomenon took place when foreign travel restrictions lead to situation that very few individuals traveled abroad (in fact, only one person went to Croatia and another person to Norway). Indeed, most individuals reported visits to the Baltic coast or lake districts. As holiday leave is typically linked with outdoor activities, outdoor activity during holiday leave in Poland is usually enough to obtain vitamin D concentration above 20 ng/mL, with some individuals even surpassing a 30 ng/mL cut-off. Our data are consistent with another study performed in the Polish population from Poznan [35]. A 12-week Nordic walking exercise resulted in a doubling of vitamin D concentrations during a March-June period. The extent of vitamin D increase was also dependent on the holiday-related duration of sun exposure among healthy office workers in Sydney (Australia) [36]. Interestingly, mean end-of-summer vitamin D concentrations in this population (27.2 ± 10.8 ng/mL) were also below a 30 ng/mL cut-off.

Though this was not our study’s primary goal, we have also assessed the prevalence of severe vitamin D deficiency (i.e., concentrations below 10 ng/mL). Prevalence of severe vitamin D deficiency was 9.8% in spring and was slightly lower than the prevalence observed by Braczkowski et al. [37], who diagnosed severe vitamin D deficiency in 16.7% of primary school children (classes IV-VI). We note, however, the crucial importance of timing for testing for severe vitamin D deficiency, as in our study cohort, it was present in one individual only (0.8%) in early autumn. Such massive discrepancy between severe vitamin D deficiency (20.2% vs. 0.1%, for March vs. October) was also demonstrated in a study by Chlebna-Sokół et al. [38]. Considering such huge differences in the prevalence of severe vitamin D deficiency raises a question about the optimal timing of such testing. It appears that testing in spring overestimates the prevalence of severe vitamin deficiency. Given that patients with severe vitamin D deficiency are more likely to be referred for further investigations, such as coeliac screen, etc., there is a clear need to standardise testing protocols to avoid unnecessary investigations implicating additional health service costs.

## 5. Conclusions

In summary, our study demonstrates that the majority of healthy individuals from a single general practice fail to reach a 30 ng/mL vitamin D cut-off. In our opinion, this raises the question as to the validity of such cut-off and whether it should be quoted as a desirable vitamin D sufficiency target. On the other hand, at least a two-week holiday leave in Poland is usually sufficient to ensure a vitamin D sufficiency status without extra supplementation as long as a 20 ng/mL cut-off is applied. We also note huge season-related discrepancies in the prevalence of severe vitamin D deficiency, so a clear consensus is needed to standardise the optimal period for testing.

## Figures and Tables

**Figure 1 ijerph-18-08116-f001:**
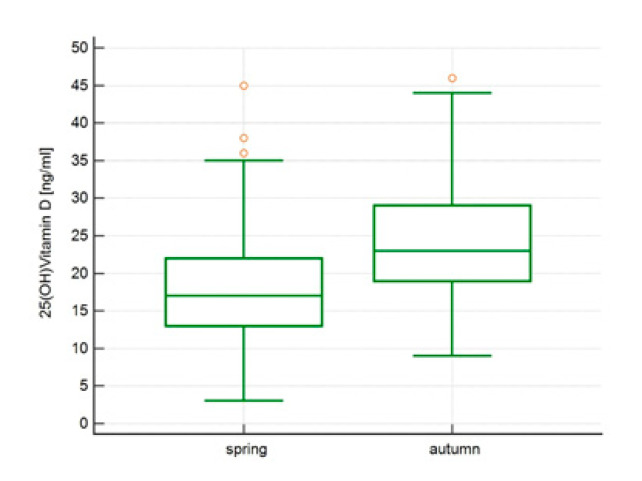
Vitamin D concentrations [ng/mL] in spring vs autumn.

**Figure 2 ijerph-18-08116-f002:**
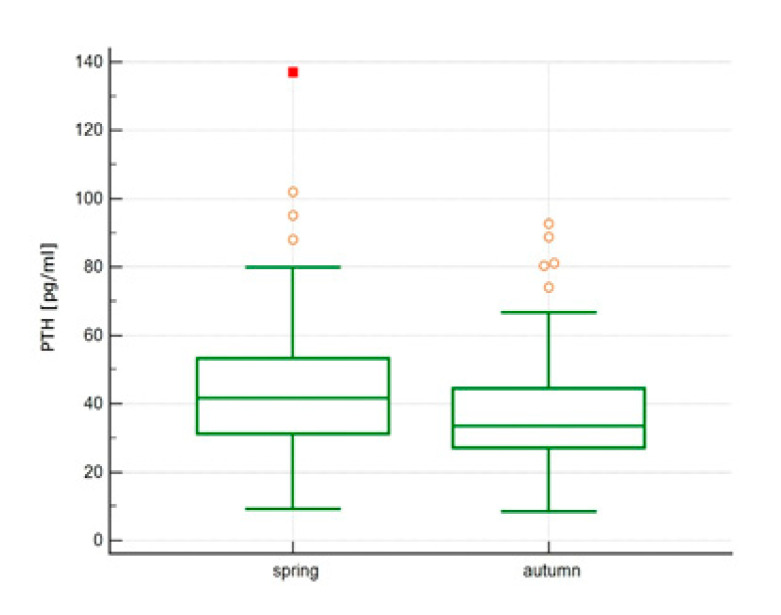
Parathormone (PTH) concentrations [pg/mL] in spring vs autumn.

**Figure 3 ijerph-18-08116-f003:**
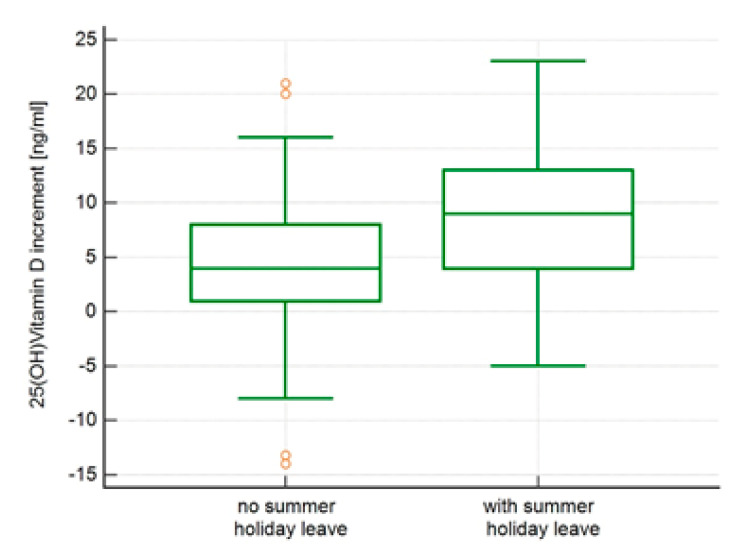
Comparison of vitamin D increments [ng/mL] between individuals who had at last two-week period of summer holiday leave (Mann-Whitney U-test, *p* = 0.005).

**Table 1 ijerph-18-08116-t001:** Prevalence of vitamin D sufficiency, insufficiency, deficiency, and severe deficiency depending on the accepted cut-offs for spring vs. autumn, *p* < 0.001, chi^2^ test.

	25(OH)D	
	Spring (*n* 132)	Autumn (*n* 125)
Definition 1		
Sufficiency > 20 ng/mL (%)	45 (34.1)	83 (66.4)
Insufficiency 12–20 ng/mL (%)	64 (48.5)	39 (31.2)
Deficiency < 12 ng/mL (%)	23 (17.4)	3 (2.4)
Definition 2		
Sufficiency > 30 ng/mL (%)	7 (5.3)	29 (23.2)
Insufficiency 20–30 ng/mL (%)	43 (32.6)	59 (47.2)
Deficiency < 20 ng/mL (%)	82 (62.1)	37 (29.6)
Severe deficiency < 10 ng/mL (%)	13 (9.8)	1 (0.8)

**Table 2 ijerph-18-08116-t002:** Correlation analysis of the measured parameters (Spearman rank method), for all subjects, for women and men. Statistically significant (*p*< 0.05) variables are highlighted in bold font.

		25(OH)D	PTH	Calcium	BMI
Whole group (*n* 132)	PTH	**−0.208**	**-**	**-**	**-**
	Calcium	0.0658	**−0.257**	-	-
	BMI	−0.0504	**0.202**	**−0.358**	-
	Age	0.0126	**0.614**	**−0.464**	**0.614**
Women (*n* 76)	PTH	−0.129	**-**	**-**	**-**
	Calcium	−0.129	−0.215	-	-
	BMI	0.0279	**0.263**	**−0.361**	-
	Age	−0.00676	**0.406**	**−0.447**	**0.547**
Men (*n* 56)	PTH	−0.167	-	**-**	**-**
	Calcium	0.156	**−0.462**	-	-
	BMI	**−0.339**	**0.255**	**−0.406**	-
	Age	−0.156	0.200	**−0.519**	**0.711**

**Table 3 ijerph-18-08116-t003:** Increments in vitamin D concentrations stratified according to having at least two-week holiday leave. The difference is statistically significant (*p* = 0.005).

Vitamin D Increment (ng/mL)	No Holiday Leave	No Holiday Leave
*n*	58	67
max	21.00	23.02
Median	4.01	9.03
95% CI for the median	2.0003 to 6.0004	2.0003 to 6.0004

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
