# Peer review of "Effect of Summer Sunshine Exposure on Vitamin D Status in Young and Middle Age Poles: Is 30 ng/mL Vitamin D Cut-Off Really Suitable for the Polish Population?"

_ijerph, 2021, doi:10.3390/ijerph18158116_

Round 1

Reviewer 1 Report

This is a very well written paper, presenting an interesting study in a very clear way on the usefulness and definition of cut-off point for 25(OH)D. I enjoyed reading it.

I have some minor comments:

L72: legibility can be improved by not only reporting the number of males, but also the females. Please add that. Furthermore, how was the age distribution among the males and females and their individual BMI and was this more or less equally divided? Please add this information as a separate table either in the Methods sections or as supplementary information to enable the readers to assess the soundness of your sample. In addition, why was the age range of 6-50 years chosen? Please elaborate on the choices that were made concerning this.

L76: Why were the repeated measurements only taken from 125 individuals, instead of the 132 at the start? Please clarify.

Table 2: at the male section, the authors forgot to put a dot at the calcium line.

L160-162: What does the holiday leave entail? After Figure 3, you explain the holiday leaves, but I think it would be better to put this part of text above Figure 3. Otherwise, readers may wonder to where the individuals went on holidays.

L192: " A study on large number', please add 'a' between 'on' and 'large'.

Author Response

We thank the Reviewer for useful and kind comments and enclose our reply below:

L72: legibility can be improved by not only reporting the number of males, but also the females. Please add that. Furthermore, how was the age distribution among the males and females and their individual BMI and was this more or less equally divided? Please add this information as a separate table either in the Methods sections or as supplementary information to enable the readers to assess the soundness of your sample. In addition, why was the age range of 6-50 years chosen? Please elaborate on the choices that were made concerning this.

We included these data in the Materials & Methods section – lines 70-74, than now includes age and BMI demographics data for men and women.

As explained in the Introduction (lines 53-61, references 11, 12), the lowest vitamin D concentrations appear to be present in teenage or young population (20-39 years of age), where ingestion of vitamin D supplements is negligible. On the other hand, at least in Poland, individuals above 50 (and females in particular) tend to take vitamin D supplements as osteoporosis prevention that might have caused significant bias for our observations, that included only “supplement-free” individuals. On the other end of spectrum vitamin D supplementation is recommended for children during their vaccination period, that typically coincides with early childhood and pre-school years. In order to clarify this, we have added an extra sentences to the Materials & Methods section (lines 75-80).

L76: Why were the repeated measurements only taken from 125 individuals, instead of the 132 at the start? Please clarify.

Each study is characterized by some drop-out rate, i.e. in this case some individuals failed to attend their second appointment. We believe, however, that data should be presented in their genuine form, even if this highlights some imperfections. In our opinion this drop-out rate (7/132, i.e. 5.3%) does not affect the validity of our observations.

Table 2: at the male section, the authors forgot to put a dot at the calcium line.

This has been corrected.

L160-162: What does the holiday leave entail? After Figure 3, you explain the holiday leaves, but I think it would be better to put this part of text above Figure 3. Otherwise, readers may wonder to where the individuals went on holidays.

We explained this in the Results section above Figure 3, lines 171-173.

L192: " A study on large number', please add 'a' between 'on' and 'large'.

This has been corrected.

Reviewer 2 Report

The paper submitted is written very well and is very timely regarding the issue of appropriate vitamin D blood levels for sufficiency.  Its results can be extrapolated far beyond the limited subject group in Poland and generalized to other populations.

Most of my critique relates to format issues:

l. 120-121 must be placed on the preceding page of the manuscript

Table 2: I suggest instead of using italics to identify statistical significance, that the authors bolden those data that are significant.\

Table 2 is split over 2 pages and it should appear on one page or the other.

l. 181-2 should be moved to the subsequent page to appear with the rest of Table 3.

After line 78, a reference should be added which will give the reader the approximate amount of vitamin D that would be in the food of a similar group of subjects. 

Author Response

We thank the Reviewer for useful and kind comments and enclose our reply below

l. 120-121 must be placed on the preceding page of the manuscript

This has been corrected.

Table 2: I suggest instead of using italics to identify statistical significance, that the authors bolden those data that are significant.

Thank you for this useful comment, we have indeed changed italics into bold fond and it looks more legible.

Table 2 is split over 2 pages and it should appear on one page or the other.

This has been corrected.

l. 181-2 should be moved to the subsequent page to appear with the rest of Table 3.

This has been corrected.

After line 78, a reference should be added which will give the reader the approximate amount of vitamin D that would be in the food of a similar group of subjects. 

We quote the recommended vitamin D intake (at least 10 ug/day) – now reference 13. There are unfortunately no good Polish population data on average vitamin D intake in this age group (6-50 years). Available Polish data pertain mostly to people with metabolic syndrome (obese, average age 57 years, working in agriculture, i.e. not urban population) (Godala M et al. Med Pr. 2021 Feb 3;72(1):9-18. doi: 10.13075/mp.5893.01021) or children only (Chlebna-Sokól B, Blaszczyk A Med Wieku Rozwoj. 2003 Apr-Jun;7(2):173-80). We have found, however, a study performed on a population from Greater Manchester – Kift R et al. Int. J. Environ. Res. Public Health 2018, 15, 1624; doi:10.3390/ijerph15081624 (age 12-60 with subdivision into a purely Caucasian subpopulation), where average vitamin D intake was much lower, i.e. about 4 µg/day and added this as reference 14, lines 88-92.